# Collaborative Analysis on the Marked Ages of Rice Wines by Electronic Tongue and Nose based on Different Feature Data Sets

**DOI:** 10.3390/s20041065

**Published:** 2020-02-15

**Authors:** Huihui Zhang, Wenqing Shao, Shanshan Qiu, Jun Wang, Zhenbo Wei

**Affiliations:** 1Department of Biosystems Engineering, Zhejiang University, 866 Yuhangtang Road, Hangzhou 310058, China; 21813008@zju.edu.cn (H.Z.); n21913051@zju.edu.cn (W.S.); jwang@zju.edu.cn (J.W.); 2College of Materials and Environmental Engineering, Hangzhou Dianzi University, 1158 Baiyang Street, Hangzhou 310018, China; Qiuss@hdu.edu.cn

**Keywords:** e-tongue, e-nose, rice wine, modified electrodes, smartphone

## Abstract

Aroma and taste are the most important attributes of alcoholic beverages. In the study, the self-developed electronic tongue (e-tongue) and electronic nose (e-nose) were used for evaluating the marked ages of rice wines. Six types of feature data sets (e-tongue data set, e-nose data set, direct-fusion data set, weighted-fusion data set, optimized direct-fusion data set, and optimized weighted-fusion data set) were used for identifying rice wines with different wine ages. Pearson coefficient analysis and variance inflation factor (VIF) analysis were used to optimize the fusion matrixes by removing the multicollinear information. Two types of discrimination methods (principal component analysis (PCA) and locality preserving projections (LPP)) were used for classifying rice wines, and LPP performed better than PCA in the discrimination work. The best result was obtained by LPP based on the weighted-fusion data set, and all the samples could be classified clearly in the LPP plot. Therefore, the weighted-fusion data were used as independent variables of partial least squares regression, extreme learning machine, and support vector machines (LIBSVM) for evaluating wine ages, respectively. All the methods performed well with good prediction results, and LIBSVM presented the best correlation coefficient (R^2^ ≥ 0.9998).

## 1. Introduction

Chinese rice wine, which is made from glutinous rice, wheat Qu, and yeast, is one of the most popular alcoholic beverages in China and East Asian [1]. The fermentation and aging process can enrich the flavor profile of Chinese rice wine. Empirically, aging time has great impact on the quality of rice wines, and the oxidative process can provide rice wine with modification of the olfactory and gustatory characteristics. Therefore, the aging wines have better quality, and customers prefer to choose rice wine with long aging time [2,3]. Some unscrupulous merchants labelled the spurious aging information on the packages for profiteering, and those fake and inferior wines harm the interests of consumers and disturb the market seriously. As the traditional method, a sensory panel is commonly used for the detection of wine ages. However, sensory evaluation requires experienced experts and reliable evaluation criterions [4,5,6,7], and it is also influenced by physiological and psychological conditions of experts. Precision instruments are also used for the identification of wine ages by analyzing the chemical components, such as gas chromatography with mass spectrometry (GC-MS) [8], ultra-performance liquid chromatography (UPLC) [9], high-performance liquid chromatography coupled with mass spectrometry (HPLC-MS) [10,11], and UV-Vis spectrophotometry [12]. In general, those large-scale and expensive instruments need complex preparation procedures. Moreover, the flavoring substances cannot be detected completely by those instruments, and the flavor profile cannot be accurately reproduced. Therefore, it is significant to find a simple, fast, and efficient method to identify the wine ages.

Electronic tongue (e-tongue) and electronic nose (e-nose) offer a fast, low cost, and non-destructive alternative to detecting samples without sample preparation, [13,14,15]. E-tongue (e-nose) can simulate human gustatory (olfactory) evaluation process to evaluate food qualities via ‘taste’ (smell) information [16,17,18,19], and both the instrument has been used successfully for evaluating the wine quality [20,21,22,23,24]. Although the e-tongue and e-nose were not used for detecting the quality of Chinese rice wine, the successful usage of the instruments indicated that e-tongue and e-nose have technique potential for identifying the wine age of rice wines. Sensor array is the kernel part of the instruments, and it can be used for obtaining the flavor fingerprint information of food products [25,26,27]. As one of the most important working parameters of the sensor array, sensitivity influences the obtained flavoring information directly. The concentrations of the main flavoring substances in rice wine (such as glucose (Glu), tyrosine (Tyr), ascorbic acid (AA), isobutanol, and ammonia) are very low, and the sensor arrays of those commercial e-tongue and e-nose are not sensitive enough for obtaining complete flavor information from the rice wines. Therefore, a novel e-tongue and e-nose were self-developed by The Agricultural Equipment and Intelligent Detection Team (AE and ID) of Zhejiang University.

Different concentrations of the flavoring substances in the rice wines with different ages lead to the diversity of flavors. Tyr, AA, and Glu are the typical taste substances, which have high correlations with wine ages [28]. In this work, the developed e-tongue included three types of conductive polymers/noble metal nanoparticles (CPs/NMNPs) nanocomposite-modified electrodes which exhibited high sensitivity to Tyr, AA, and Glu, respectively. The nanocomposites can enlarge the surface area of electrodes and provide more active sites for catalyst [29,30]. Two types of chronoamperometry (multi-frequency staircase pulse voltammetry (MSPV) and multi-frequency rectangle pulse voltammetry (MRPV)) were used for obtaining taste information of wine samples. The e-nose was equipped with a smartphone and presented as more portable and intelligent. A smartphone application (APP) was self-compiled for operating the e-nose, and it also could be used for displaying the smell signal curves. All the olfactory information obtained by the e-nose was stored in a cloud platform.

Flavor of rice wine consists of taste and smell information; both types of information play crucial roles and the application of one apparatus (e-nose or e-tongue) is insufficient to obtain the complete flavor profile of Chinese rice wines [31]. Multi-sensor data fusion technology is defined as the comprehensive decision-making process, and the merging of the data set with different types can probably improve the efficiency of the identification models [32,33]. In this work, the flavor information was obtained by self-developed e-tongue and e-nose, and the information was fused to improve the accuracy of identification results. Six types of feature data sets (included four fusion data set) were established, and Pearson coefficient analysis and variance inflation factor (VIF) analysis were used for optimizing the fusion matrixes by removing the multicollinear information. The combination of electronic nose and tongue used for identifying the rice wine has never been reported. The main purposes of this study were: (1) To investigate the most efficient method for fusing the response data obtained by e-nose and e-tongue; (2) to investigate whether the wine ages can be classified and predicted accurately by the fusion data based on pattern recognition methods.

## 2. Materials and Methods

### 2.1. Reagents and Apparatus

Sulfanilic acid (ABSA), acid chrome blue K (ACBK), aspartic acid (ASP), Glu, Tyr, and AA were obtained from Aladdin Chemical Co. Ltd. (Shanghai, China). HAuCl_4_ and H_2_PtCl_6_ were purchased from Sinopharm Chemical Reagent Co. Ltd., China. The bare glass carbon electrodes (GCE) were purchased from IDA Co. Ltd. (Tianjin, China). The solution of 0.1 M Na_2_HPO_4_ · 12H_2_O and 0.1 M NaH_2_PO_4_ · 2H_2_O were mixed as the 0.1 M Phosphate buffer solution (PBS). All the solutions were prepared with deionized water, and all the chemicals applied in the study were of analytical grade.

The FE^2^-FiveEasy Plus pH meter (METTLER TOLEDO, Greifensee, Switzerland) was used for the measurement of pH values. Electrochemical experiments were performed by using PARSTAT 3000A electrochemical workstation (AMETEK. Inc. Berwyn, USA). A standard three-electrode configuration included a saturated Ag/AgCl reference electrode (3.5 M saturated KCl, diameter 2 mm), a platinum auxiliary electrode (diameter 2 mm), and three nanocomposites modified working electrode. The BS224S electronic analytical balance (Sartorius, Goettingen, Germany) was used for weighting the chemical substances. The electrodes were cleaned by a SK1200H ultrasonic cleaner (KUDOS, Shanghai, China). The surface morphologies of the modified electrodes were presented by the SU8010 scanning electron microscope (SEM, HITACHI, Tokyo, Japan). Also, 50 mL electrochemical cells and 250 mL beakers were prepared for the sampling experiment of e-tongue and e-nose, respectively.

### 2.2. Sample Preparation

Guyuelongshan rice wines (Shaoxing city, Zhejiang province) of 3, 5, 8, 10, and 20 years (GY-3Y, GY-5Y, GY-8Y, GY-10Y, and GY-20Y) were bought from a local supermarket and stored at 4 °C, and 200 samples (40 samples of each marked age) were randomly selected for the e-tongue and e-nose experiment. All the wine samples were stored at room temperature one night before measurements.

### 2.3. The Electronic Tongue Based on Conducting Polymer Nanocomposites Modified Electrodes

#### 2.3.1. Preparation of the Conducting Polymer Nanocomposites Modified Electrodes

Three types of modified electrodes (PASP/PtNPs/GCE, PABSA/AuNPs/GCE, and PACBK/AuNPs/GCE), which have high sensitivities to Tyr, AA, and Glu, were self-fabricated. Each of the GCEs was polished with 0.3 and 0.05 μm alumina powder before modification, and then rinsed thoroughly by using ultrasonic cleaner with deionized water, nitric acid solution (V/V = 1:1), and absolute ethyl alcohol.

PASP/PtNPs/GCE was prepared by electrochemically depositing the PASP polymer onto the surface of bare GCE in the mixture of 2 mM PASP and 0.1 M PBS (pH 6.0). Cyclic voltammetry was applied in three-electrode cell between the potential range of −1.2 V to + 2.0 V (scan rate 100 mV s^−1^) for 15 cycles. The PASP/GCE was submitted to 24 successive potential sweeps from −0.25 to 0.4 V with a scan rate of 50 mVs^−1^ in the mixed solution of 0.5 M H_2_SO_4_ and 6 mM H_2_PtCl_6_.

The sequences of preparation process for PABSA/Au/GCE were quite different. The GCE was placed in 1.2 mM HAuCl_4_ solution containing 0.1 M KNO_3_ for 240 s at a fixed potential of −0.2 V, so that the gold nanoparticles were deposited on the GCE surface. Then, the Au/GCE was inserted into a mixture of 2 mM sulfanilic acid and 0.1 M PBS (pH 7.0) and scanned by the cycling potential from −1.5 to + 2.5 V at a scan rate of 100 mV s^−1^ for 12 cycles.

In order to fabricate the PACBK/AuNPs/GCE, the mixture solution was prepared by adding 0.5 mM ACBK into 0.1 M PBS (pH 6.0). Then, the bare GCE was immersed into the mixed solution with the scanning potential ranged from −1.6 V to + 2.0 V. The freshly acquired PACBK/GCE was electrodeposited in 1.2 mM HAuCl_4_ solution (containing 0.1 M KNO_3_) for 240 s at a fixed potential of −0.2 V.

Finally, all the fabricated electrodes were washed repeatedly with de-ionized water to remove the electrolyte and monomer, and then naturally dried at room temperature and kept in reserve.

#### 2.3.2. The Experimental Procedures of E-Tongue

The rice wine samples were detected by e-tongue through a standard three-electrode configuration at room temperature (Figure 1). Each of the modified electrodes were taken as the working electrode, in turn; a saturated Ag/AgCl and a platinum wire were used as the reference electrode and auxiliary electrode, respectively. The standard solutions of Tyr, AA, and Glu were prepared to investigate the sensitivities and limits of detection (LOD) of each modified electrode, and the relationship between the peak currents and solution concentrations was explored by electrochemical methods. Then, the fabricated modified electrodes were used for detecting wine samples (50 mL of each sample, 40 samples for each wine age).

Two types of samples were prepared before the experiment: (1) The wine sample mixed with 10 mL 0.1 M PBS (pH 2.0) was prepared for the detection of PACBK/AuNPs/GCE and PABAS/AuNPs/GCE; (2) the mixture containing wine sample and 10 mL 0.05 mol/L NaOH solution was prepared for the detection of PASP/AuNPs/GCE. After each modified electrode was dipped into its corresponding solution, MRPV and MSPV were used as the scanning potential to obtain the taste information. Three replicates were carried out for each sample and the average values were taken as the final detection result. The adsorbate on the electrode surfaces was rinsed by acidic solution (2 M of HNO_3_, 10 mM of H_2_O_2_) and de-ionized water. The detection of each sample followed the same procedures.

### 2.4. The Portable Electronic Nose Equipped with Smart Phone

#### 2.4.1. The Development of the Portable E-nose with Smart Phone

The self-developed e-nose consisted of gas sensor array, control module, power module, wireless communication module, human–computer interface, and cloud storage module (Figure 2a). The gas sensor array included 12 Metal-Oxide-Semiconductor (MOS) sensors which were highly sensitive to rice wines. The brief description of primary attribute and detection range of each MOS sensor are listed in Table 1. All those sensors were placed properly in a white rectangular Teflon chamber, and the inner structure of the sensor chamber was optimized by ANSYS Fluent Software (ANSYS, Inc., Canonsburg, PA, USA) to ensure each sensor met the VOCs efficiently. An Android application was self-compiled as the human–machine interface to control all operations and display the response signals, procedure parameters, and discrimination results. Aliyun cloud storage platform (Alibaba Inc., Hangzhou, China) was used for storing responses data, working parameters, and identification models.

As shown in Figure 2b,c, the typical response curves were obtained by the e-nose from different samples. The gas fingerprint information obtained from the different samples was significantly different: The signals obtained from the 20-year sample ranged from 0 to 3000 mv, and the signals obtained from the 5-year sample ranged from 0 to 3500 mv. Otherwise, the response signals obtained by the S4 and S1 sensor showed the maximal and minimal values in Figure 2b (the signals obtained from the 20-year sample), respectively. The response signals obtained by the S11 and S6 sensor showed the maximal and minimal values in Figure 2c (the signals obtained from the 5-year sample), respectively. During the aging time, alcohols and aldehydes can be changed into organic acid by catalytic and oxidation reaction, and the esterification reaction of alcohols accelerates in the aging process of rice wine. Meanwhile, the concentration of ethyl alcohol becomes lower because ethyl alcohol molecules are associated with water molecules by hydrogen bonds. Therefore, the volatile organic compounds (VOCs) of rice wine change during the aging process, which can affect the response values obtained by e-nose from those rice wine samples. It was obvious that the signals obtained from the 20-year and 5-year sample were different, and those differences can be used for the identification of rice wines.

#### 2.4.2. The Experimental Procedures of E-Nose

The detection of VOCs information was applied by the portable e-nose. Two hundred samples (40 samples of each marked age) were random selected in the experiment, and 50 mL of each sample was put into a 250 mL gas blanketing flask. Rice wine is a highly volatile product at room temperature, and 10 min is enough for the headspace-generation. The volatile components of rice wines were pumped from the headspace of flask into the sensors chamber with the intake rate of 300 mL/min, then the volatile components attached to the surface of the sensors and caused variation of the sensor resistance. The response curves changed violently at the beginning of the detection phase and then showed stable tendency from the 10th to the 50th second. The sensors was cleaned from the 50th to the 230th second with clean-dry-air at a constant rate of 100 mL/min to remove the coated substrate on the surfaces of MOS sensors, and the responses in the cleaning phase were also obtained as the after taste information to investigate the desorption ability of each sensor. The olfactory information (both the taste and aftertaste information) contained the characteristics of wine samples. All the information was uploaded to the Aliyun cloud platform and used for the identification works based on different pattern recognition methods.

### 2.5. Data Fusion Techniques

Multi-sensor data fusion technology is defined as the comprehensive decision-making process based on the information obtained by multiple sensors [34,35,36]. There are two problems with the data fusion: (1) How to efficiently fuse the response signals obtained by different types of sensors, and (2) how to ensure the fused information present better performance than the single usage of e-nose or e-tongue. Multi-sensor information can be fused by original-set fusion, feature-set fusion, and decision-set fusion [37,38], and the feature-set fusion method was applied in the study.

The feature data of the two types of original responses were extracted by area method (the area under the corresponding response curves). Each feature data set was normalized, and then merged together for establishing a new fusion data set. Because of the unbalanced number of features between e-tongue (with 18 features) and e-nose (with 24 features), the feature data obtained by e-nose played a dominant role in the identification model. Actually, the weight of the gustatory information was higher than the weight of the olfactory information in the sensory evaluation of rice wine. According to GB/T13662-2008 <rice wine> 6.1 “sensory test”, the sensory evaluation of rice wine was based on appearance, aroma, taste, and flavor, i.e., factor set X = {x_1_, …, x_i_, …, x_n_} = {appearance, aroma, taste, flavor}. The weight of each factor was obtained according to the GB/T13662-2008: W = {W_1_, …, W_j_, …, W_n_} = {0.10, 0.30, 0.40, 0.20} [39]. Two methods can be used for establishing the weighted data set. The first method was that the number of feature dimensionality of e-nose reduced to the number of the feature dimensionality of e-tongue. However, dimension reduction is often accompanied by the loss of the information. The second method weighted the standardized e-tongue features to enlarge the numeric range, and no information was lost during the weighted process. Therefore, the second method was used for the study, and the weighted features of the e-tongue were fused directly with the standardized features of e-nose. This weighted-fusion data set was analyzed in the follow-up works. Moreover, the multicollinearity among separate sensors hindered the establishment of regression models, and the feature data with high correction might lead to incorrect discrimination and prediction [40]. In the study, Pearson coefficient analysis and variance inflation factor (VIF) analysis were used for optimizing the fusion matrixes by decreasing the multicollinear information, and the total four new fusion data sets (direct-fusion data set, weighted-fusion data set, optimized direct-fusion data set, and optimized weighted-fusion data set) were analyzed in follow-up studies.

### 2.6. Pattern Recognition Methods

In this work, five pattern recognition methods were used for analyzing the feature data sets. Principal component analysis (PCA) and locality preserving projections (LPP) were performed for classifying the wine samples. Partial least squares regression (PLSR), extreme learning machine (ELM), and support vector machines (LIBSVM) were performed as regression methods. Minitab 16 (Pennsylvania State University, USA) was used for building PLSR model, and MATLAB R2017b software (The Math-Works Inc., USA) was used for building other models.

PCA is a linear dimensionality reduction approach. It can convert multidimensional data into a linearly independent variables matrix and the variables are arranged according to the contribution rates (called principal components (PCs)) [41,42]. Generally, the first two (PC1 and PC2) or three (PC1, PC2, and PC3) columns of data could explain most of the variables. PCA tends to be used as a tool to observe data structures.

LPP is a classical linear unsupervised technique, the same as traditional PCA, so it could be considered as a further promotion algorithm [43]. Based on LPP, only the local information, data points, and the adjacent points are employed to obtain lower dimensional features, which could avoid divergence of the data set and maintain a good local neighbor relationship just like the original information.

PLSR, a statistical method to model a response variable, is the combination of the PCA and other multivariate linear regression analysis [44]. It is excellent for highly correlated or even collinear predictors and application to the regression modeling analysis.

ELM is a feed forward neural network raised by professor Guang-Bin Huang [45], with a single input layer, a hidden layer, and an output layer. In most cases, the input weights and hidden nodes are randomly assigned, and the output weights are directly calculated by the least square method in just a single step. The optimized output result will be one, and only when the parameters of hidden nodes are confirmed.

Based on statistical learning theory, LIBSVM was developed at the National Taiwan University and presented as a library for SVM, supporting discrimination and regression [46]. There are relatively few adjustable parameters involved in the procedure compared with SVM.

## 3. Results and Discussion

### 3.1. The Gustatory Information Obtained by Electronic Tongue

#### 3.1.1. Physicochemical Characterization of Modified Electrodes

Structural morphologies of the PASP/PtNPs, PABSA/AuNPs, and PACBK/AuNPs were investigated by SEM. As shown in Figure 3a, the gold nanoparticles were successfully deposited onto the surfaces of the bare electrodes, but those particles were not uniform size and some particles were grouped together (Figure 3a). The conglomeration of NPs could have led to the decrease of surface-to-volume ratio and the poor stability. The morphology of PtNPs film exhibited an obvious three-dimensional structure, which could be beneficial to maintain large electroactive area (Figure 3b). The micrograph of the nanocomposition of PABSA/AuNPs, PACBK/AuNPs, and PASP/PtNPs is shown in Figure 3c–e. The bright and furry features were the organic polymers, which formed a homogeneous layer and strong bounded the NPs. The CPs attached on the NPs surfaces induced colloid repulsion among metallic nanoparticles, which can prevent the aggregation among each metallic nanoparticle. Moreover, the CPs materials could have improved the electrochemical characteristics of the nanocomposite-modified electrodes.

The catalytic performance of those modified electrodes to those taste substances (Glu, Tyr, and AA) were presented by using cyclic voltammetry (CV). It was obvious that no catalytic peak was presented on the surface of bare GCE (Figure. 3f). A well-defined anodic peak potential of Tyr was presented on the surface of AuNPs/GCE (0.65 V) and PABSA/AuNPs/GCE (0.67 V), respectively, and the catalytic peak potential of AuNPs/GCE shifted more negatively. As shown in Figure 3g, the oxidation peak of AA on the surface of PACBK/AuNPs/GCE (12.38 μA) was about 6 times as much as the oxidation peak current of the bare GCE (2.08 μA) and 1.7 times as much as that of the AuNPs/GCE (10.22 μA). PASP/PtNPs/GCE presented the maximum oxidation peak current at −0.32 V, which was larger than the oxidation peak current of bare GCE and PtNPs/GCE (Figure 3h). All these results obviously indicated that the conductive polymer nanocomposites can accelerate the electron transfer rates and increase the peak currents because of its larger specific surface area and remarkable synergistic effect.

#### 3.1.2. Responses Presentation and Feature Data Extraction

In this work, the three types of modified electrodes constituted the sensor array of e-tongue for the identification of rice wines with different ages. As shown in Table 2, the concentration of Tyr, AA, and Glu in rice wine are higher than the LOD of each substance (Table 2). Due to the disturbances of other chemicals, the exact concentration of Tyr, AA, and Glu in the rice wine might be difficult to detect, but the electrochemical behaviors of the taste substances on the electrodes surface could be obtained. Those electrocatalysis and electrooxidation information had high associations with the aging characteristic of rice wines.

Two types of potential waveforms (MRPV and MSPV) were used for the modified electrodes as the scanning waveforms (Figure 4a,b), and each potential waveform included three frequency phases (1 Hz, 10 Hz, and 100 Hz). MRPV worked 12.21 s (11 s, 1.1 s, 0.11 s for each frequency phase) and MRPV worked 7.77 s (7 s, 0.7 s, 0.07 s for each frequency phase) for detecting one wine sample. Therefore, (11 s + 1.1 s + 0.11 s)/0.02 s × 3 electrodes = 18,315 data and (7 s + 0.7 s + 0.07 s)/0.02 s × 3 electrodes = 11,655 data were obtained by MRPV and MSPV during the experiment, respectively. However, most of the data had high correlation with each other, and they were useless for building a pattern recognition technique. Therefore, it was necessary to extract feature data from origin response.

In previous studies, the maximum response values, maximum integral values, maximum differential values, and the maximum slop values were always taken as the feature data [47,48]. The complete flavor information of the sample could not be presented by just one feature, and the most useful information could be abandoned during the feature extracting process. In the study, the area method was used for extracting the feature data from the original signals obtained by MRPV and MSPV (Figure 4c,d). Thousands of original data can be reduced to six feature data (3 electrodes × 2 potential waveforms = 6 data). In the following analysis, those feature data (the data matrix of 200 samples × 6 features) were used as the input variables of pattern recognition methods.

### 3.2. The Olfactory Information Obtained by Electronic Nose

#### 3.2.1. The Responses Information Obtained by E-nose from Wine Samples

The portable e-nose system was also used for detecting rice wine samples, and the specialized smart phone APP was compiled to display the response curves of 12 MOS sensors. As shown in Figure 5a, the typical response curve obtained by S10 sensor (TGS2603) from GY-3Y is presented. S10 sensor responded quickly after adsorbed volatile organic compounds. The signal curve went up rapidly and reached its maximum value (about 2200 mV) after 10 s, and the signal kept the value to the 50th s. Then, a slight cleaning process was initiated with clean-dry-air at a constant rate of 100 mL/min, and the signal curve decreased continuously. The whole experimental stage was over at the 230th second. The response tendency of each sensor was similar to S10, just different in the response values. All the signal data obtained by 12 MOS sensors constitute the olfactory information of rice wine samples.

#### 3.2.2. Feature Data Extraction

The area method was also used for extracting the feature data from the e-nose responses. As shown in Figure 5b, the area features extracted from one response curve included two parts: (1) The sum of areas under the response curve from 0 to the 50th second, which can be regarded as the taste information; and (2) the sum of areas under the response curve from the 50th to the 230th second, it can be regarded as the after taste information. The taste and aftertaste information were taken as the feature data to build the pattern recognition algorithm, respectively. Therefore, total 24 features (12 sensors × 2 features = 24 features) were obtained from each rice wine sample, and the features data matrix [200 × 24] of the wine samples was taken as the input variables of algorithms for the further identification analysis.

### 3.3. Discrimination of Rice Wine Samples Based on E-Tongue, E-Nose, and Fusion System

#### 3.3.1. The Discrimination Results Based on the Single Usage of E-Tongue

In this work, an e-tongue based on PCA and LPP was used for visualizing the features data and for qualitative classifying wine samples with different marked ages. As shown in Figure 6a, the sum of PC1 and PC2 manifested 89.56% of the variance. The five group samples of different ages (GY-3Y, GY-5Y, GY-8Y, GY-10Y, and GY-20Y) were marked with different colors and shapes, and all the samples were separated from each other clearly. Compared with other samples, GY-3Y samples were not well grouped. It could be noticed that the PC1 value of each group sample decreased along with the increase of marked ages, except for the PC1 values of GY-10Y. The LPP result of the five types of wines are presented in Figure 6b. The discrimination results based on LPP were similar to the PCA result, and all the samples were classified completely. LPP1 values of GY-3Y, GY-5Y, GY-8Y, and GY-20Y were increased along with the increasing marked ages, which showed the opposite tendency of PCA result. In general, it could be concluded that the e-tongue based on conducting polymer nanocomposites modified electrodes had the potential to discriminate rice wines.

#### 3.3.2. The Discrimination Results Based on the Single Usage of E-Nose

As presented in Figure 6c, the feature data obtained by e-nose from rice wines are visualized in the PCA plot. The first two principal components (PC_1_ and PC_2_) were enough to explain the total 79.92% variance of the data sets. The five groups of rice wine cannot be classified clearly in the PCA plot, and the GY-5Y samples overlapped with the GY-8Y seriously. Nonetheless, the samples from the same category gathered closely, and the PC1 values of GY-Y20, GY-Y10, GY-Y8, GY-Y5, and GY-Y3 went up along the X axis from left to right. Figure 6d shows that LPP performed better than PCA. Each type of sample was more concentrated and the distance between each group increased. However, samples of different ages still overlapped each other, especially the samples of GY-3Y and GY-5Y. The LPP1 values of GY-3Y, GY-5Y, GY-8Y, GY-10Y, and GY-20Y, which also presented similar location tendency, were enlarged along with the increase of marked ages. Although the e-nose performed worse than e-tongue in the discrimination work, the results demonstrated the potential of e-nose in identifying rice wines with different ages.

#### 3.3.3. The Discrimination Results Based on the Combination of E-Tongue and E-Nose

In the study, the flavor information obtained by e-tongue and e-nose were fused in feature level, and 42 original features (18 feature data obtained by e-tongue and 24 feature data obtained by e-nose) for each sample built a 200 × 42 data set (200 samples × 42 features). Two types of the feature-set fusion method (the direct feature fusion and weighted feature fusion method) were applied in the study, and discrimination results based on each data set were compared with each other by using PCA and LPP.

The direct-feature fusion set (200 samples × 42 features) was built by merging the 18 feature data of e-tongue with 24 feature data of e-nose. As shown in Figure 7a, the first two PCs (PC1 and PC2) explained 72.18% of the variance. Five groups of rice wines could not be separated clearly, and the groups of GY-5Y, GY-8Y, and GY-20Y were close to each other. Figure 7b shows that LPP performs more powerful than PCA, and each group can be classified completely. The discrimination results of LPP based on the direct-fusion data set also worked much better than the former LPP result based on the single usage of e-tongue or e-nose.

The weighted-feature fusion set was built by weighting e-tongue features with a parameter W (W = 16/9), and the new data matrix was still 200 × 42. As described in Figure 7c, the discrimination results of the five rice wine groups based on the weighted-feature fusion set were clearer than that based on the direct-feature fusion set, and all five group samples could be classified completely. LPP also performed much better in the discrimination of rice wines based on the weighted-fusion data set (Figure 7d). Each type of sample was grouped closely and classified clearly with each other.

Because the response signals were obtained from the same instrument, either the direct-fusion or the weighted-fusion feature matrixes had a high degree of multicollinearity. E1a1–E3b3 were obtained from e-tongue, and S1a1–S12a2 were obtained from e-nose. The absolute Pearson values ranged from 0 to 1 on behalf of low and high multicollinearity, respectively. As shown in Figure 8, the correlations among feature data were visualized by using Pearson correlation matrix, and the features with higher Pearson value had more redundant information. The Pearson value between E3a2 and E1a3 (the feature data obtained by e-tongue based on different electrodes) was smaller than the Pearson value between E3a2 and E3a3 (the feature data obtained by e-tongue based on the same electrode). Those two fusion data sets presented similar multicollinearity (Figure 8a,b). The visualization results revealed the probability of misclassification among the five group samples of rice wines. Therefore, a filtration criteria (limit value of VIF was set 10) was used for optimizing feature matrixes by eliminating the redundant information, and the feature corresponding to the maximum VIF value was excluded until all the features of VIF were less than 10 [49,50,51]. It can be seen from Figure 8c that the multicollinearity almost disappeared after the optimization based on VIF method. As shown in Figure 9, both the direct-fusion and weighted-fusion worked better than the optimized data set in the discrimination, although all the samples could be classified completely. The location of rice wine samples presented more diffuse distribution, and each group was located very close with each other. As discussed above, PCA and LPP based on the optimized data sets did not presented the best discrimination results. It might be the reason that the optimized fusion data sets lost some information by removing some features, which affected the working efficiency of the discrimination models.

Comparing the discrimination results based on different fusion methods, the LPP with weighted-fusion data set proved the most efficient method in the discrimination work. Aroma and taste are the most important attributes of rice wines. The gustatory information obtained by e-tongue and the olfactory information obtained by e-nose composed the complete flavor information of rice wine. Moreover, the weights of smell and taste are different for the sensory panel, and the correction values based on the weights can improve the discrimination results of rice wines. Therefore, the weighted-fusion data set had the highest correlation with rice wines, and the wine ages of rice wine can be identified completely by the combination of e-nose and e-tongue based on the weighted responses. However, the wine ages could not be identified from the LPP plots directly. Although the LPP1 values of GY-3Y, GY-5Y, GY-8Y, and GY-20Y increased along the X axis from left to right, the GY-10Y samples performed irregular performance. Therefore, the regression methods based on the weighted-fusion data set were used for the following prediction works of the wine ages.

### 3.4. Prediction of Wine Ages by using the Weighted-Fusion Data Based on PLSR, ELM, and LIBSVM

In the study, the weighted-fusion data set was taken as the input data of PLSR, ELM, and LIBSVM for the prediction of the wine ages.

Overall, 200 samples were used for the prediction work. 125 samples (25 samples of each type) were taken as the training set, and 75 samples (15 samples for each group) were taken as the testing set. Radial basis function (RBF) was taken as the core function of LIBSVM, and the best parameters for ELM (n = 27) and LIBSVM (c = 1024 and g = 0.0313) were applied in the regression works. The squared correlation coefficients (R^2^) and mean square error (MSE) of both the training and testing set were calculated to evaluate the performances of the PLSR, ELM, and LIBSVM regression models, and higher R^2^ with lower MSE indicated the best regression model. As shown in Table 3, all the methods performed well for the prediction work with the weighted-fusion data set. LIBSVM presented the best results, the R^2^ and MSE value of the testing set was R^2^ = 0.9998 and MSE = 0.0077, respectively. PLSR presented a little bit worse value with the testing set (R^2^ = 0.9941, MSE = 0.2385), but it also indicated the ability of the combined usage of e-nose and e-tongue. The prediction results of the ELM method (R^2^ = 0.9997, MSE = 0.0113) based on the testing set were better than the results of PLSR and worse than the results of LIBSVM. Therefore, LIBSVM performed much stably based on either the training set or testing set, and both the correlation coefficients (R^2^) of the training and testing sets were over 0.9998.

## 4. Conclusions

In the study, the developed e-tongue and e-nose based on five pattern recognition methods (PCA, LPP, PLSR, ELM, and LIBSVM) were used for the identification of rice wine of different wine ages (GY-3Y, GY-5Y, GY-8Y, GY-10Y, and GY-20Y). Six types of features data sets (e-tongue data set, e-nose data set, direct-fusion data set, weighted-fusion data set, optimized direct-fusion data set, and optimized weighted-fusion data set) were evaluated for the identification works. The novel e-tongue based on three nanocomposite-modified electrodes was a more powerful tool than the portable e-nose for the identification of Chinese rice wines. The fusion data set was optimized by removing the redundant features. However, those multicollinearity signals still contained discrepant information, and the original weighted-fusion data set performed better than the optimized data. The PLSR, ELM, and LIBSVM were used for predicting the wine ages based on the weighted-fusion data, and both the training and testing sets of LIBSVM presented the best results. Therefore, rice wines with different ages can be well identified by the combination of the self-developed e-tongue and e-nose based on different pattern recognition method.

Meanwhile, some limitations affected the identification results of rice wines. The number of the electronic tongue sensors was few, and the sensitivity and selectivity of each sensor could be improved. The antijamming capability of the electronic nose was a little weak, especially in a complex environment. Therefore, a further study will be done to improve the hardware and software of e-tongue and e-nose, and more efforts will be made to monitor the flavor changes of rice wines during the aging process.

## Figures and Tables

**Figure 1 sensors-20-01065-f001:**
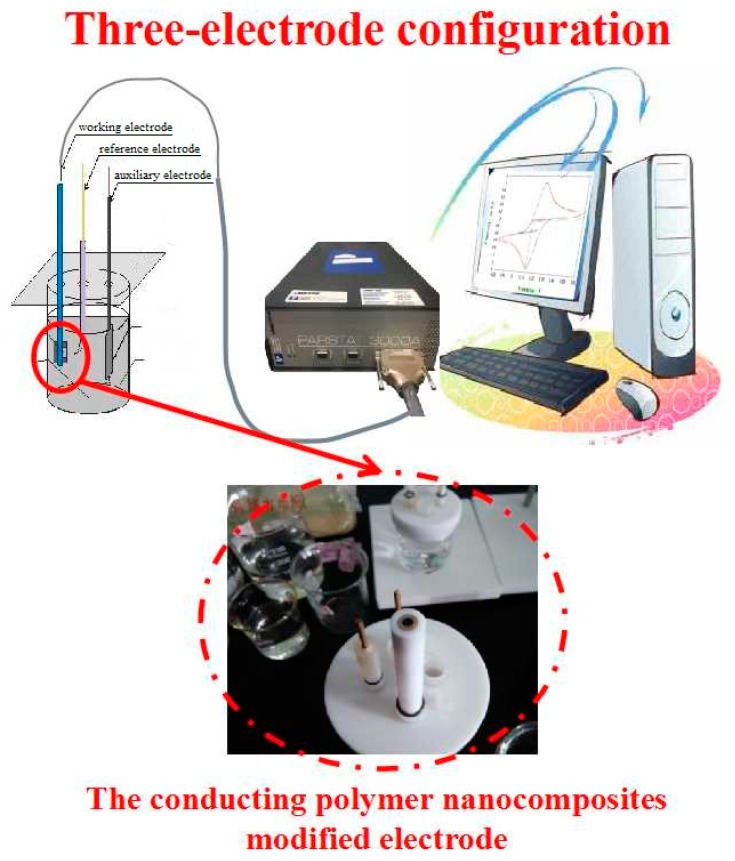
The set-up of the electronic tongue.

**Figure 2 sensors-20-01065-f002:**
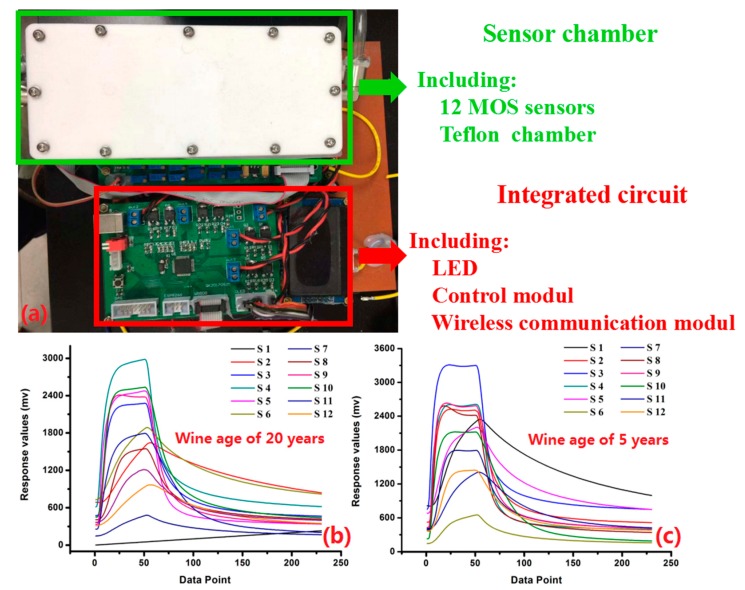
The set-up of the portable electronic nose (**a**) and the typical responses of 20-year (**b**) and 5-year (**c**) samples.

**Figure 3 sensors-20-01065-f003:**
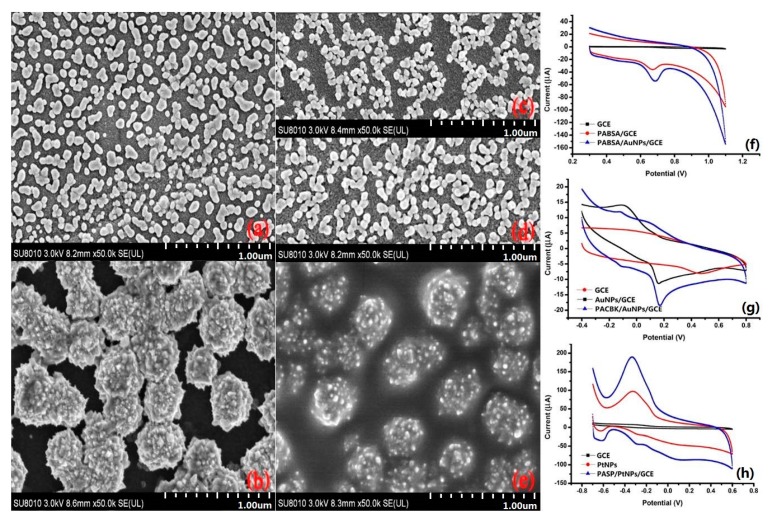
SEM images of AuNPs film (**a**), PtNPs film (**b**), PABSA/AuNPs film(**c**), PACBK/AuNPs film (**d**), and PASP/PtNPsfilm (**e**); CV behaviors of Tyr (**f**), AA (**g**), and Glu (**h**) on the raw GCE and modified electrodes.

**Figure 4 sensors-20-01065-f004:**
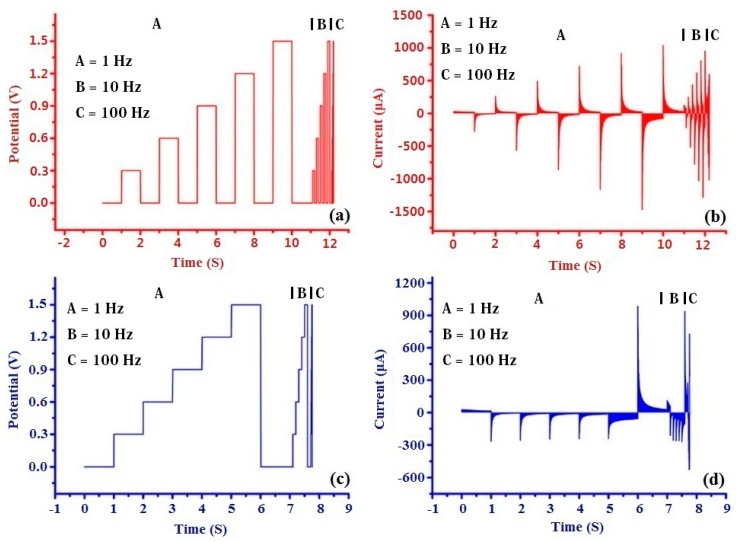
The applied potential waveforms of multi-frequency rectangle pulse voltammetry (MRPV) (**a**) and multi-frequency staircase pulse voltammetry (MSPV) (**c**); the response curves obtained by PABSA/AuNPs/GCE based on MRPV (**b**) and MSPV (**d**).

**Figure 5 sensors-20-01065-f005:**
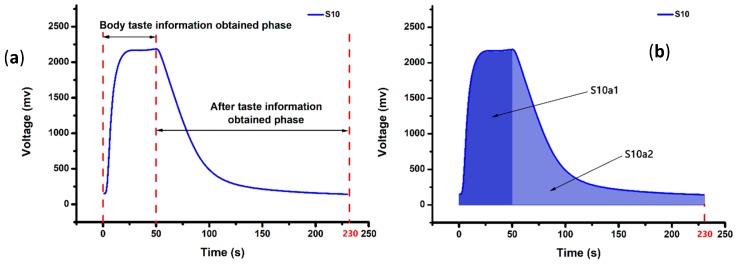
(**a**) The response curve obtained by S10 from GY-3Y wine sample. (**b**) The area under the corresponding response curve: 0~50th second taken as S10a1 and the rest as S10a2.

**Figure 6 sensors-20-01065-f006:**
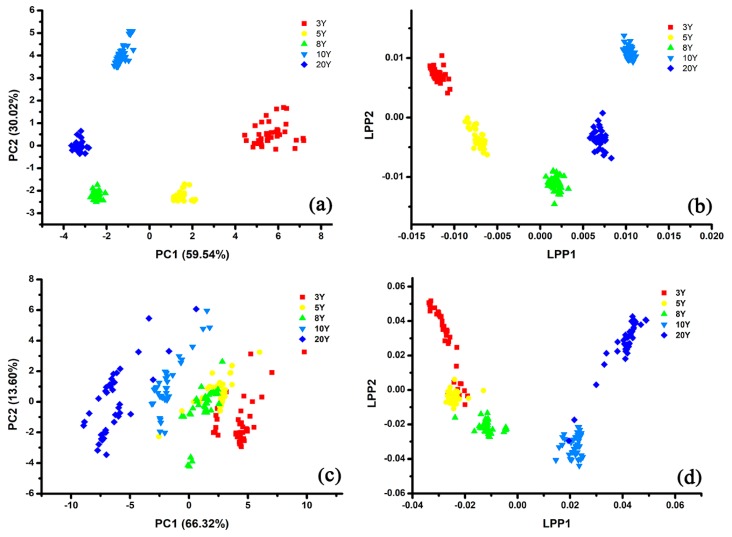
PCA and locality preserving projections (LPP) plots of rice wine samples with different ages (GY-3Y, GY-5Y, GY-8Y, GY-10Y, and GY-20Y) based on original feature data ((**a**,**b**) for e-tongue, (**c**,**d**) for e-nose).

**Figure 7 sensors-20-01065-f007:**
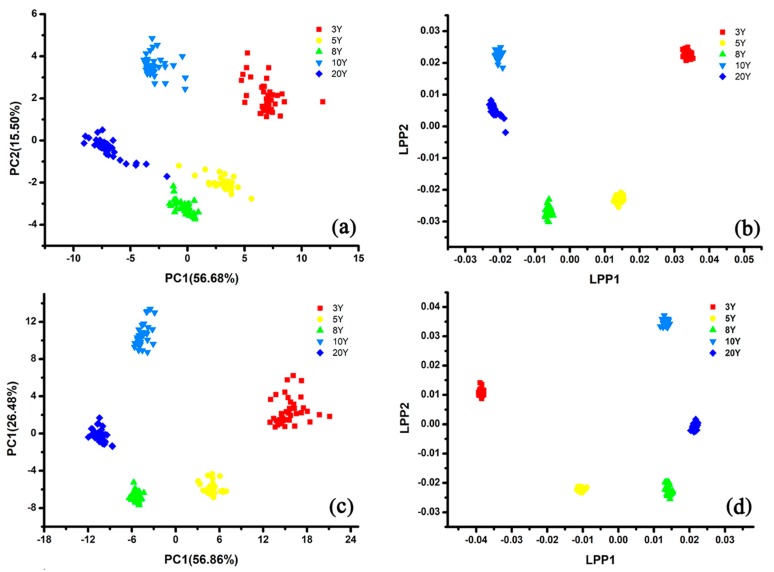
PCA and LPP score plots of the area features data sets obtained from four fusion methods: (**a**,**b**) Direct-fusion data set, (**c**,**d**) weighted-fusion data set.

**Figure 8 sensors-20-01065-f008:**
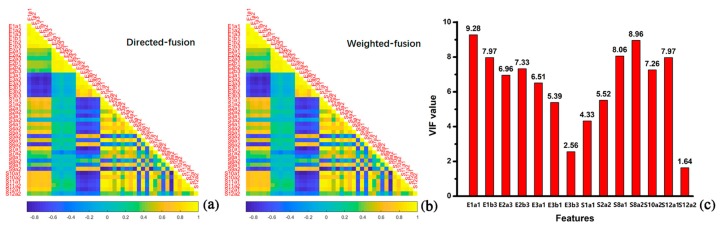
Visualization of the correlations between sensor signals based on Pearson correlation matrix: (**a**) Direct-fusion data set; (**b**) weighted-fusion data set. Plot of variance inflation factor (VIF) values calculated for the optimized characteristic data sets after direct-fusion and weighted-fusion (**c**).

**Figure 9 sensors-20-01065-f009:**
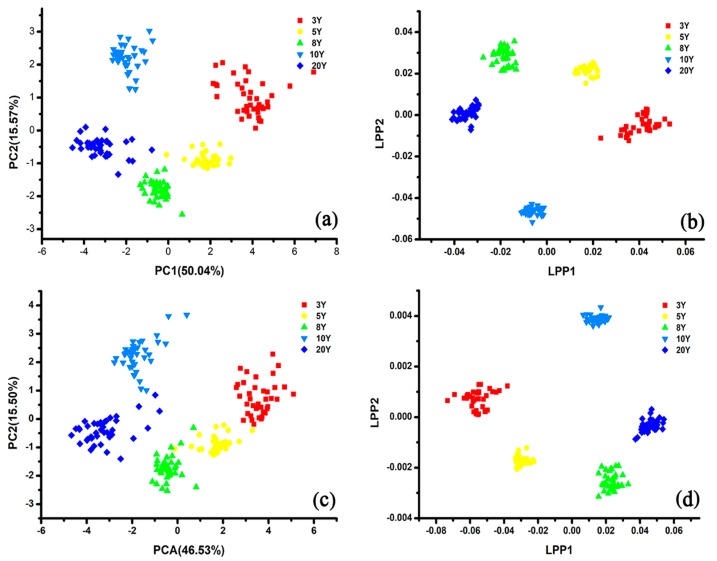
PCA and LPP score plots of the area features data sets obtained from four fusion methods: (**a**,**b**) Optimized direct-fusion data set, (**c**,**d**) optimized weighted-fusion data.

**Table 1 sensors-20-01065-t001:** The 12 MOS Sensors of e-Nose and Their Performance.

Sensors	Labels	Prime Attributes	Detection Range
TGS826	S1	Ammonia	30–300 ppm
TGS822	S2	Alcohol, Solvent vapors	50–5000 ppm
TGS816	S3	Methane, Butane, Propane	500–10,000 ppm
TGS813	S4	Methane, Butane, Propane	500–10,000 ppm
MQ138	S5	Aldehydes, Alcohols, Ketones, Aromatics	1–100 ppm
MQ137	S6	Ammonia	5–500 ppm
TGS2620	S7	Alcohol, Solvent vapors	50–5000 ppm
TGS2611	S8	Methane	1–25%
TGS2610	S9	Butane, Propane	1–25%
TGS2603	S10	Trimethylamine, Methyl mercaptan	1–10 ppm
TGS2602	S11	Organic vapors, Ammonia, Hydrogen sulfide	1–30 ppm
TGS2600	S12	Hydrogen, Ethanol	1–30 ppm

**Table 2 sensors-20-01065-t002:** The Concentrations and Limits of Detection (LOD) of Tyr, AA, and Glu.

	Tyr	AA	Glu
The concentration in rice wines	70.0–1979.5mg/L	5.7–43.2 mg/L	1552.0–2978.0 mg/L
LOD of Corresponding electrode	0.07 mg/L	0.35 mg/L	0.36 mg/L

**Table 3 sensors-20-01065-t003:** Comparison of Partial Least Squares Regression (PLSR), Extreme Learning Machine (ELM), and Support Vector Machines (LIBSVM) Models Based on the Weighted-Fusion Approach.

Technique	Algorithm	Training			Testing	
		R^2^	MSE		R^2^	MSE
Features fusion	PLSR	0.9945	0.1935		0.9941	0.2385
	ELM	0.9998	0.0089		0.9997	0.0113
	LIBSVM	0.9999	0.0053		0.9998	0.0077

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
