# Peer review of "Collaborative Analysis on the Marked Ages of Rice Wines by Electronic Tongue and Nose based on Different Feature Data Sets"

_sensors, 2020, doi:10.3390/s20041065_

Round 1

Reviewer 1 Report

Comments:

The authors proposed a sensing system to detect the aroma and taste of rice wine. The system is divided into two sections. One is gas sensing part, where they used commercial different gas sensors as a gas sensor array which is connected different control modules to collect the information from the gas sensor array and observe the sensing performance by smartphone, they also applied different algorithm to identify different aroma/gas present in the rice wine. There is no novelty of this section because the authors didn’t fabricate/design the gas sensors or signal processing system, they only used commercial different gas sensors as well as modules. The authors already published similar work previously on the journal named “Sensors”, doi:10.3390/s17112500. In the second section is taste sensor, the authors proposed taste sensor array using three types of modified electrodes to detect glucose, tyrosine, and ascorbic acid in rice wine. The authors already published similar work previously in the journal named “Sensors and Actuators B”, 255 (2018) 895–906. http://dx.doi.org/10.1016/j.snb.2017.08.155

There is no novelty of the present work. The present work is the combination of their previously published work.

Below presents the copy of  the part picture of their present (Fig.1) and published (Fig.1 Sensors 2017, 17, 2500doi:10.3390/s17112500.) work .

Author Response

The authors proposed a sensing system to detect the aroma and taste of rice wine. The system is divided into two sections. One is gas sensing part, where they used commercial different gas sensors as a gas sensor array which is connected different control modules to collect the information from the gas sensor array and observe the sensing performance by smartphone, they also applied different algorithm to identify different aroma/gas present in the rice wine. There is no novelty of this section because the authors didn’t fabricate/design the gas sensors or signal processing system, they only used commercial different gas sensors as well as modules.

Thank you for the comment and thank you for providing a chance to explain the novelty of the e-nose. The details were as follows:

nose is a complex detection system, it include sensor array, data collection module and signal processing system. I want to explain the novelty of the developed-e-nose from two parts: sensors array and signal processing system.(1) Sensors array. As we know, there are few types of commercial e-nose applied for the food quality detection. However, the sensor array of those commercial e-nose are not chose for one certain food samples (to maximize the application of the e-nose). Therefore, the sensors and their working parameters may not very suitable for the rice wine samples. We have detected the VOCs of rice wine with GC-IMS, and chose the best suitable sensor to compose the sensor array of the e-nose based on the GC-IMS results. We also tested the optimal working parameters of the sensor array (such as signal time, clean time, temperature, etc) for the detection of rice wine. Moreover, we have designed the sensor chamber based on the Fluent software(ANSYS, Inc), and the optimized inner structure of the sensor chamber  made sure each sensor could touch the gas evenly. The selection of sensor array, optimization of working parameters, and the designment of sensor chamber for the detection of rice wine have not been reported before. (2) Signal processing system. The identification results of the e-nose are effected serious by the selected feature data and pattern recognition methods. We have self-compiled the signal processing system by using C++ programming language, and the signal processing system can run smoothly in the smartphone (we have applied for the software copyright, 2019SR1139174). Based on the signal processing system, we can chose the specific feature data from the original signals for the identification of rice wine. The redundant information and interference information also could be removed from the feature data by the signal processing system. Then, the pattern recognition method (PCA, SVM, ANN, etc) based on the optimized feature data was applied for the identification works, and the working parameters of the pattern recognition method could be automatic optimized for the classification and prediction of rice wine samples. The signal processing system with smartphone applied for the identification of rice wine have not been reported before.

    The developed e-nose is the specific system for the identification of rice wine. Each part of the e-nose, such as sensor array, data collection module and signal processing system, was designed for the specific detection of rice wine. And the specific e-nose applied for the detection of rice wine have not been reported.  

The authors already published similar work previously on the journal named “Sensors”, doi:10.3390/s17112500.

Thank you for the comment. There were much different between the two papers, and the explanation was as follows:

The published paper (DOI: 10.3390/s17112500) was about the development of a e-nose for detecting the quality of rice wine based on the olfactory information. In the research, we developed our first version potable e-nose. Although the results were good for the identification of the rice wine with different wine ages, all the samples could not be classified clearly in the PCA, LLE or LDA plot. For improving the identification results, we have checked the e-nose and analyzed the results carefully and two main conclusions was obtained. (1) Both the aroma and taste are the most important attributes of rice wines, the olfactory information obtained by single application of e-nose may not enough for the classification of rice wine. We have looked through many Government Standards about how to evaluate the wine quality (such as the beer, red wine, liqueur, etc) based on sensory panel, each of the Government Standard mentioned that the weight of taste is important as that of aroma. Therefore, the addition of the gustatory  information obtained by e-tongue might improve the classification results. In the research, we fused the information obtained by e-nose and e-tongue, and four types of fusing data sets were applied for the classification and prediction of rice wine (directed-fusion data set, weighted-fusion data set, optimized directed-fusion data set and optimized weighted-fusion data set). All the rice wine samples based on the fusion data set could be classified clearly, and the additional gustatory  information improved the analysis results exactly. (2) The redundant and multicollinear information obtained by the e-nose might lead to the bad classification results. The sensors of the e-nose were not sensitive to one certain substance, they have cross-sensitive to the substances own similar structure. Moreover, the variation of environmental parameters (such as temperature, humidity, gas, etc) could influence the e-nose signals. Therefore, the redundant and multicollinear information was unavoidable obtained by the e-nose. In the published paper (DOI: 10.3390/s17112500), the original data were not optimized by removing the redundant and multicollinear information, and were applied as the input data of those pattern recognition methods directly. In the research, the obtained features were optimized by pearson coefficient analysis and variance inflation factor (VIF) analysis, and two types of feature data sets (optimized directed-fusion data set and optimized weighted-fusion data set) were applied for the identification analysis.

In the research, we focus on effective fusing the e-nose and e-tongue data and removing the redundant and multicollinear information from the original data. They have much different from the published paper (DOI: 10.3390/s17112500). We have also changed the title of the submitted paper as “Collaborative analysis on the marked ages of rice wines by electronic tongue and nose based on different feature data sets” to   highlight the theme of the research and prevent misunderstanding between the published paper and the submitted paper.     

In the second section is taste sensor, the authors proposed taste sensor array using three types of modified electrodes to detect glucose, tyrosine, and ascorbic acid in rice wine. The authors already published similar work previously in the journal named “Sensors and Actuators B”, 255 (2018) 895–http://dx.doi.org/10.1016/j.snb.2017.08.155

There is no novelty of the present work. The present work is the combination of their

previously published work.

Thank you for the comment. We did not combine the previously published work, many new analyses were added in the research. The explanation was as follows:

The published paper (DOI: 10.1016/j.snb.2017.08.155) was about the development of a sensor array of e-tongue for detecting the quality of rice wine. As I have explained in the second part, both the aroma and taste are the most important attributes of rice wines, the gustatory information obtained by single application of e-tongue may not enough for the classification of rice wine. We have combined the e-nose and e-tongue to analysis the tea quality (DOI: 10.1039/c5ra17978e), and the good classification and prediction results proved that the combination system worked better than the single usage of e-nose or e-tongue. However, the universal commercial e-nose (Airsense Company, German) and e-tongue (Alpha MOS Company, France) was applied in the research.

We already self-developed the specific e-nose and e-tongue system for identification of rice wines, and we wanted to test the efficiency of the combination of self-developed e-nose and e-tongue for the rice wine analysis. Therefore, we focus on the optimization of feature data and fusion of the e-nose and e-tongue data in the research. The classification results based on the six types of data sets (e-tongue data set, e-nose data set, directed-fusion data set, weighted-fusion data set, optimized directed-fusion data set and optimized weighted-fusion data set) were compared with each other, and the obtained features were optimized by pearson coefficient analysis and variance inflation factor (VIF) analysis. The classification results indicated that the gustatory information among rice wines with different ages contributed more than the olfactory information. Therefore, the submitted paper was much different with the published paper (DOI: 10.1016/j.snb.2017.08.155).       

Below presents the copy of a picture of their present and published work.

     Thank you for the comment. The e-nose presented in the published paper (DOI: 10.3390/s17112500) was the First Version. We already improved the hardware, and the Second Version presented higher integration. The Second Version of the e-nose was already presented in the paper, and the picture of the Second Version was as follows:

Reviewer 2 Report

Please, carefully revise the English grammar and typo errors. (e.g. page 1, lines 13, 15, 16; page 3, l. 106, 124; page 4, l. 182; and so on).

PCA and LPP are techniques employed for discrimination tasks, and not for classification. Please correct it.

How do the authors control the dimensionality of voltammograms data if CVs were scanned in different range potentials?

I suggest writing: “3, 5, 8, 10 and 20 years” instead of “3 years, 5 years, 8 years, 10 years, and 20 years”.

Please explain how and why the wine samples were stored for one night before measurements.

Do the LPP provide score values or any quantitative parameter?

Authors show only e-nose sensor scheme, why the e-tongue was not depicted?

In my point of view only LPP from Fig. 5 can discriminate the rice wine samples. Please improve the discussion to make clear the innovation and advantages of the proposed method. It would be interesting you also show in table 3 the MSE of directed-fusion and separated e-tongue and e-nose data set.

Please add the MRPV, MSPV and e-nose response signal response in the Support Information.

Please add scale bar in Figure 2a-e.

I suggest renaming the Figures 2 – CV – a-c (f-g in the text) as Figures 3a-c. The text is different from legend of Figure 2. Please check it.

Why the films were changed for each analyte (Tyr, AA, Glu)?

Correct the legend of Figure 3.

Please increase the readability of Figure 7.

Please explain the meaning of absolute value of Pearson between -1 and 0.

How the VIF values were calculated? Please explain the reason to limit VIF value to 10. Add more references.

How long did the experiments lead to this kind of analysis? 

Is the number of electrodes really necessary (12 from e-nose and 9 from e-tongue)? In case of sensor substitution is necessary a new calibration?

Reviewer 3 Report

Is it 'potable' or 'portable' electronic nose? All 'potable' should be replaced by 'portable' in the manuscript.

Introduction must be improved. It should include more recent references, trends, currently known solutions, and research findings about the development of such technologies and sensors for wine age detection/identification. In addition, the state-of-art of the sensors should be clearly presented.

Present your novelty or impact to the field in the Introduction section.

Explain clearly about the selectivity and stability or reproducibility of the sensors and devices used. Also about their recovery or response times. Describe about the reliability of the TGS gas sensors used in terms of future gas sensing and other applications. It is advised to cite the following references: - https://doi.org/10.3390/bios6040060  - https://doi.org/10.1007/s12161-016-0739-4            - https://doi.org/10.1007/s12161-013-9778-2 - https://doi.org/10.3390/proceedings2131061  - https://doi.org/10.1016/j.snb.2005.03.078 - https://doi.org/10.1016/j.proeng.2016.11.126

Add limitations and disadvantages of both the systems developed in Discussion section.

English language and grammatical errors must be revised throughout the manuscript. The quality of Figure 3 must be improved. Make the font sizes uniform throughout the manuscript. Explain about the feasibility of such a system developed for practical applications. What about the power consumption?

Round 2

Reviewer 2 Report

The authors have clarified the reviewer's questions and now the article can be considered for publication.

About query 9: Actually, I suggested to present all the response curves obtained by e-tongue (for the 9 electrodes) and e-nose (for the 12 electrodes) of each wine during the experiments and add them in SI. Because Figure 4 and 5 are the response curves due to only one sensor and one analyte. It would be interesting to show the differences obtained for the other sensors and analytes.

Author Response

About query 9: Actually, I suggested to present all the response curves obtained by e-tongue (for the 9 electrodes) and e-nose (for the 12 electrodes) of each wine during the experiments and add them in SI. Because Figure 4 and 5 are the response curves due to only one sensor and one analyte. It would be interesting to show the differences obtained for the other sensors and analytes.

Thank you for the comment. All the response curves obtained by e-tongue and e-noseof each wine during the experiments and add them in SI.

Reviewer 3 Report

The authors have revised the manuscript as per the comments provided. The paper is deemed to be published after some grammatical errors check.

Author Response

The authors have revised the manuscript as per the comments provided. The paper is deemed to be published after some grammatical errors check.

Thank you for the comment. I have looked through the manuscript carefully, and the english language and grammatical errors were revised throughout the manuscript. The revised parts were marked by red in the manuscript.
